# Recommendation of SLM Process Parameters Based on Analytic Hierarchy Process and Weighted Particle Swarm Optimization for High-Temperature Alloys

**DOI:** 10.3390/ma16165656

**Published:** 2023-08-17

**Authors:** Ze-Jun Zhang, Yuan-Jie Wu, Ze-Ming Wang, Xiao-Yuan Ji, Wei Guo, Dong-Jian Peng, Xian-Meng Tu, Sheng-Zhi Zhou, Huan-Qing Yang, Jian-Xin Zhou

**Affiliations:** 1State Key Laboratory of Materials Processing and Die & Mould Technology, School of Materials Science and Engineering, Huazhong University of Science and Technology, Wuhan 430074, China; m202270979@hust.edu.cn (Z.-J.Z.); 18384258727@163.com (Y.-J.W.); npicwzm@126.com (Z.-M.W.); weiguo@hust.edu.cn (W.G.); d202180438@hust.edu.cn (X.-M.T.); zhousz22@163.com (S.-Z.Z.); zhoujianxin@hust.edu.cn (J.-X.Z.); 2Xi’an Space Engine Company Limited, Xi’an 710100, China; pdjian29@163.com (D.-J.P.); yhuanqing@163.com (H.-Q.Y.)

**Keywords:** selective laser melting, process recommendation, particle swarm optimization, analytic hierarchy process, high-temperature alloy

## Abstract

Selective laser melting (SLM) of high-temperature alloys involves intricate interdependencies among key process parameters, such as laser power and scanning speed, affecting properties such as density and tensile strength. However, relying solely on experiential knowledge for process parameter design often hampers the precise attainment of target requirements. To address this challenge, we propose an innovative approach that integrates the analytic hierarchy process (AHP) and weighted particle swarm optimization (WPSO) to recommend SLM process parameters for high-temperature alloy fabrication. Our proposed AHP–WPSO model consists of three main steps. First, a comprehensive historical database is established, capturing the process parameters and performance metrics of high-temperature alloy SLM parts. Utilizing an AHP framework, we compute the performance similarity between target and historical cases, applying rational thresholds to identify analogous cases. When suitable analogs are elusive, the model seamlessly transitions to the second step. Here, the WPSO model optimizes and recommends process parameters according to target specifications. Lastly, our experimental validation of the GH4169 high-temperature alloy through SLM experiments corroborates the effectiveness of our AHP–WPSO model in making process parameter recommendations. The outcomes underscore the model’s high accuracy, attaining a recommendation precision of 99.81% and 96.32% when historical analogs are present and absent, respectively. This innovative approach offers a robust and reliable solution to the challenges posed in SLM process parameter optimization for high-temperature alloy applications.

## 1. Introduction

Additive manufacturing (AM), a revolutionary manufacturing technique, emerged in the late 1980s and has gained widespread popularity [1]. Among additive manufacturing technologies, selective laser melting (SLM) is one of the most important branches. The fundamental principle of selective laser melting (SLM) involves slicing the structure of a part using computer-aided design software, based on a digital model file. This process utilizes powder metal, enabling the creation of objects through a “bottom-up” layer-by-layer accumulation bonding approach. Due to its distinctive layered processing, this technique is widely known as 3D printing [2,3,4].

SLM technology enables the manufacturing of fully dense parts and extremely complex geometrical structures. Despite strict input processing requirements, which involve describing the characteristics of the metal powder to be used, SLM can process a variety of metallic biomaterials, including commercially pure titanium (CP-Ti), 316 L stainless steel, cobalt–chromium–molybdenum (Co-Cr-Mo), Ti-6-Al-4-V, AlSi_10_Mg, tantalum, and nickel–titanium [5]. For solid, porous, and hybrid materials, processing factors and various SLM production designs have an impact on the final mechanical characteristics of the resulting products.

Currently, there exist certain challenges in the production of SLM, including issues related to the non-standard recording of process information and the absence of crucial process design standards. Consequently, production personnel frequently rely on experiential approaches, resorting to repeated “trial and error” methods to establish new processes. This practice significantly hampers the efficiency of process research and development, leading to escalated production costs and equipment losses. As a result, there is a pressing need to undertake research on SLM process recommendations, aiming to establish a dependable foundation for subsequent SLM process designs and further enhance the efficiency of process research and development.

The meta heuristic algorithm [6] is an improvement of the heuristic algorithm, which is a combination of a random algorithm and a local search algorithm. Meta heuristic algorithms are proposed to be comparable to optimization algorithms, which can provide a feasible solution (though not necessarily an optimal one) to a problem at an acceptable cost (referring to computational time and space). Khosravani M R [7] proposed a case-based approach. The weight of different characteristics was determined by historical fault data. Rintala L. [8] combined the case-based reasoning method to establish a metallurgical process intelligent recommendation system, which can effectively guide process designs. Mohanmmed MA [9] improved case-based reasoning technology based on a genetic algorithm and applied it to the field of mobile phone fault detection, achieving a fault detection accuracy of 98.7%. The determination of reusable process design information largely relies on human experience and knowledge, requiring a large amount of manual participation, resulting in problems such as strong regularity, low universality, and low efficiency [10].

Wang Wei [11] from Huazhong University of Science and Technology proposed a case representation method of a mold NC process based on geometric semantic features and combined it with the comprehensive weighted similarity measurement method to form a recommendation for similar NC process cases. Feng Chao [12] from Northwestern Polytechnical University proposed an improved particle swarm optimization algorithm for feature weight determination and a similar case retrieval method based on an RBF neural network which could quickly form judgements and recommend an emergency decision method. Kejun Xiang [13] combined case-based reasoning with rule-based reasoning to establish a high-speed cutting database system based on hybrid reasoning. Ding Xu [14] from Nanjing University of Aeronautics and Astronautics formed process parameter recommendations and the multi-objective optimization of cutting parameters in micro-milling based on the hybrid method of case-based reasoning and rule-based reasoning. Abroad, Jiang [15] established a process planning model based on case-based reasoning for application scenarios of product remanufacturing, combined with the rough set method to reduce the dimensions of the case features, and effectively improved the product quality through the resulting process recommendation. Tung Y H et al. [16] combined rule reasoning with case-based reasoning to quickly diagnose problems in complex situations.

Most domestic and foreign scholars have conducted qualitative research on the SLM process, and the empirical rules obtained have not been transformed into high-precision quantitative models to guide subsequent process designs. As a result, the current SLM process design still relies heavily on the experience of designers, and it is difficult to accurately design SLM process schemes that meet the requirements with fewer instances of “trial and error”. Therefore, we design a hybrid system for the recommendation of SLM process parameters that can accurately recommend parameters in SLM process designs, reduce our reliance on experience, and quickly meet design performance needs.

## 2. SLM Processing Recommended Hybrid Model Construction

Selective laser melting (SLM) is a complex and dynamic high-speed process that is influenced by a variety of factors, including mechanical conditions, material characteristics, laser scanning methods, and external environmental conditions [17]. Among these factors, the most crucial ones are the process parameters, such as the laser power, scanning speed, scan spacing, and powder layer thickness. Additionally, SLM-formed parts have stringent performance requirements, including tensile strength, yield strength, density, stress distribution, and elemental composition. The stress and elemental distribution in SLM-formed components have a significant impact on their mechanical properties, reliability, and operational longevity [18,19]. However, due to the challenges in measuring stress and elemental distribution and the complexities of optimizing models based on these factors, this study employs indicators such as tensile strength, yield strength, density, and post-fracture elongation to predict the performance of the formed components.

This paper introduces an AHP–WPSO hybrid model for recommending SLM process parameters based on the existing SLM database, as shown in Figure 1. The hybrid recommendation model combines the analytic hierarchy process (AHP) with weighted particle swarm optimization (WPSO). The AHP model retrieves similar cases from the SLM database for its recommendations, aiding in process optimization and reusability. In cases in which similar target cases are absent, the WPSO model serves as a supplementary approach within the AHP framework for process optimization. This combination of methodologies facilitates the rapid generation of SLM process plans.

### 2.1. Analytic Hierarchy Process Model for Process Optimization

Case-based reasoning (CBR) is a new method of solving problems based on databases in the field of artificial intelligence. It searches for similar questions in the past in the case’s base problems and solutions to solve new problems. It is of great significance to improve the accuracy of process recommendation results by considering the weight proportion of each demand feature and combining the weight with case-based reasoning. Based on this, a weighted case-based reasoning model combined with the analytic hierarchy process (AHP) is established for the rapid optimization and reuse of process schemes. The analytic hierarchy process is used to determine the feature weight.

Firstly, the decision problem is transformed into a three-level model consisting of the target layer, the criterion layer, and the scheme layer, serving as the basis for weight allocation. For different materials, case retrieval is conducted by classifying the cases based on the material type, with independent weight allocation within each material category. Taking GH4169 as an example, the target layer represents the total weight and the criterion layer includes the weights assigned to three factors influencing the selection of SLM process schemes: the forming part characteristics, processing costs, and mechanical properties. The scheme layer comprises specific case feature weights under these three factors, including the structural features, structural size, forming time, density, tensile strength, yield strength, and elongation at break. The established case feature hierarchy model is illustrated in Figure 2.

Since criterion layer elements and scheme layer feature weights cannot be allocated across levels, judgment matrices need to be provided from top to bottom for the weight distribution, until all the case feature weights in the scheme layer are determined. The judgment matrices are constructed using a method on a scale from 1 to 9 to measure the relative importance of the indicators within each layer, as shown in Table 1.

The form of the judgment matrix is *Y*_k_ = (x_ij_)_n×n_, where k is the index of the judgment matrix, x_ij_ represents the importance scale of the pairwise performance input by the user, and n is the order of the judgment matrix. The feature matrix is a positive reciprocal matrix, meaning that x_ij_ = 1/x_ij_. Thus, the feature matrix only needs to be filled in the diagonal upper/lower part, with scales from 1 to 1/9 corresponding to 1 to 9. The horizontal and vertical coordinates of the feature importance ratio are inversely proportional.

According to the structure of the hierarchy model, there are three judgment matrices in this study: the criterion layer judgment matrix *Y*_1_, the formed part feature judgment matrix *Y*_2_, and the mechanical property judgment matrix *Y*_3_. Among them, *Y*_2_ contains only two features, and based on empirical knowledge, it is known that the importance of forming part structural features is nearly equal to that of the structural size for production. Therefore, *Y*_2_’s internal scales are directly filled with 1. In the context of aerospace and military applications, based on the practical production experience of technical personnel from a certain military unit, the demand features selected in this model are ranked in order of importance: the mechanical properties of formed parts are greater than the features of formed parts, which are greater than the processing costs. Under the uncertainty of specific service conditions for SLM formed parts, the density, tensile strength, yield strength, and elongation after fracture are equally important. The forms of the three judgment matrices are shown in Table 2, Table 3, and Table 4, respectively. After obtaining the judgment matrices, it is necessary to test their consistency to prevent significant discrepancies in the scales of the features within the same layer, which could lead to errors in the feature weights. The current approach commonly uses the consistency index (CI) to measure the degree of judgment matrix consistency, calculated as shown in Equation (1).
(1)$$CI=\frac{\lambda_{max}-n}{n-1}$$

Here, *λ_max_* represents the maximum eigenvalue of the judgment matrix, and *n* is the order of the judgment matrix.

**Table 2 materials-16-05656-t002:** Criterion layer judgment matrix.

*Y* _1_	Processing Cost	Formed Part Features	Mechanical Properties
Processing cost	1	0.5	0.2
Formed part features	2	1	0.25
Mechanical properties	5	4	1

**Table 3 materials-16-05656-t003:** Formed part feature weight judgment matrix.

*Y* _2_	Structure	Structure Size
Structure	1	1
Structure size	1	1

**Table 4 materials-16-05656-t004:** Mechanical performance weight judgment matrix.

*Y* _3_	Density	Tensile Strength	Yield Strength	Elongation after Break
Density	1	1	1	1
Tensile strength	1	1	1	1
Yield Strength	1	1	1	1
Elongation after break	1	1	1	1

A smaller consistency index indicates the better consistency of the judgment matrix. When the CI is zero, the judgment matrix is completely consistent. The average random consistency index (RI) is introduced as a reference control for matrix consistency. The RI for the orders one to ten of the positive reciprocal matrices, obtained from 1000 calculations, is shown in Table 5. When the matrix order is less than three, the judgment matrix is considered to have perfect consistency; otherwise, the consistency of the judgment matrix needs to be evaluated by comparing the consistency index (CI) with the average random consistency index (RI). Therefore, the random consistency ratio (CR) is introduced as the final indicator of judgment matrix consistency, calculated as shown in Equation (2). When the CR is less than 0.1, it is considered that the judgment matrices provided by the user have acceptable consistency, and the feature vectors serve as the weights for each level. Otherwise, the user needs to rescale the importance of the case features until passing the consistency test. After the calculation, all three judgment matrices exhibit favorable consistencies. Otherwise, users need to rescale the importance of the case features until the consistency test is passed. Based on the importance scales in the judgment matrices, the ratio of current demand feature weights is calculated using the analytic hierarchy process as follows: the forming time to density to tensile strength to yield strength to elongation after fracture to structural feature to structural size = 0.116:0.171:0.171:0.171:0.171:0.100:0.100.
(2)$$\mathrm{CR}=\frac{\mathrm{CI}}{\mathrm{RI}}$$

After calculating the weight size of the scheme layer, case retrieval and reuse are carried out based on the case library. The essence of case retrieval is to retrieve the most similar cases from the case library based on the similarity between new cases in demand and existing cases in the case library and use the solutions of similar cases as a solution to the demand. This article proposes a case retrieval method based on material classification based on the fact that there is almost no reference between different materials in the process. The main steps are as follows:

Step 1: Classify the cases in the case library by material grade. When inputting new demand cases, first search the case library by material grade and select cases of the same material to form an alternative case set; 

Step 2: Based on the weighted nearest neighbor method [21], calculate the similarity between the new case and the set of candidate cases. Essentially, after calculating the local similarity for each case feature, combined with the feature weights, summarize the weighted total similarity. The calculation formula is shown in Equation (3).
(3)$$SIM\left(A,B\right)=\overset{n}{\underset{i=1}{\sum }}sim\left(A_{i},B_{i}\right)\times w_{i}∕\overset{n}{\underset{i=1}{\sum }}w_{i}$$

Among them, *SIM*(*A*,*B*) is the weighted total similarity between new case *A* and existing case *B*, sim(*A_i_*, *B_i_*) is the local similarity between new case *A* and existing case *B* in the ith case feature, *w* is the weight of the *i*th case feature determined by the analytic hierarchy process, and *n* is the total number of case features.

From the above formula, it can be seen that the local similarity of different case features has a direct impact on the weighted total similarity of cases. Therefore, using appropriate methods to calculate the local similarity is a key step in case retrieval. The case features involved in this article can be divided into two types of data: numerical and enumeration data types. It is difficult to calculate the local similarity between numerical and enumerated font features using a unified method. Therefore, this article adopts a hybrid local similarity calculation method, as shown below:

(1) For the characteristics of numerical cases, the calculation formula [22] is shown in Equations (4) and (5).
(4)$$sim\left(A_{i},B_{i}\right)=e^{-\frac{d\left(A_{i},B_{i}\right)}{\sqrt{2}\times \sigma_{i}}}$$
(5)$$\sigma_{i}=\sigma \times \left(i_{max}-i_{min}\right)$$

Among them, the *d* (*A_i_*,*B_i_*) for the new case *A* with existing case *B*; the absolute distance between the case features is I; *σi* is the deflection point; and σ is a constant and its value ranges from 0 to 1. In this paper, it is 0.4. $i_{max}$ and $i_{min}$, respectively, represent the maximum and minimum values of the ith numerical case features. Compared with the local similarity calculation method based on absolute distance alone, the proposed method can eliminate the influence of different feature dimensions to the greatest extent and make the recommendation results more accurate.

(2) For enumerated case features, in this paper, the structural features of single finger forming parts and the local similarity are given by technicians combined with production experience. This article shows that structure characteristics are limited to eight types: ordinary block type, complex sharp angle type, impeller structure type, interlayer runner type, irregular pipe type, honeycomb structure type, complex shell type, and lattice structure type. The local similarity matrix is shown in Figure 3 and Table 6. When calculating the local similarity of structural features, please refer to the table.

After the similarity calculation, the case base sorts all the cases according to the size of the similarity. At this time, it is necessary to conduct follow-up processing for the recommended cases according to the similarity threshold S and feature weight threshold w. According to the experimental simulation results, the similarity threshold S of this model is 0.85, and the feature weight threshold w is 0.15.

### 2.2. Weighted Particle Swarm Model for Optimization of Process Parameters

Since the case itself remains unchanged, when the performance parameters required by the target exceed the maximum reasoning range of the case base, the CBR model cannot be used for process recommendations. Therefore, this paper proposes a weighted particle swarm optimization model as a supplement to the weighted case-based reasoning model, which can flexibly optimize the process parameters according to the performance requirements of the target and output the optimized process plan.

Particle swarm optimization (PSO) is a meta-heuristic population optimization algorithm for searching the global optimal solution of a problem. It was proposed by two American scholars, James Kennedy and Russell Eberhart, in 1995. Compared with other heuristic search algorithms, such as a genetic algorithm, it has the advantages of being simple and easy to realize, having fewer super parameters, being easy to adjust, and having a fast search speed and strong search ability, etc. It has achieved good results in single-objective and multi-objective optimization in engineering [25]. 

Based on the standard particle swarm optimization algorithm, this paper proposes a weighted particle swarm optimization model, as shown in Figure 4.

The construction process of the weighted particle swarm model is as follows: firstly, the optimization objective and optimization parameters are determined, then the weighted multi-objective fitness function is constructed and the constraint conditions are determined, then the particle crossing processing strategy is determined, then the superparameters are determined, and finally the conditions are optimized.

(1) The optimization objective and optimization parameter determination. The optimization objective is the input of the model, and the optimization parameter is the output of the model. Therefore, the optimization objectives are density, tensile strength, yield strength, and the elongation at break, and the optimization process parameters are the laser power, scanning speed, scanning spacing, and powder layer thickness.

(2) The construction of weighted multi-objective fitness function and the determination of constraint conditions. Since the optimization objective is multiple performance parameters, based on the established optimal performance prediction model, this paper adopts the weighted summation method to construct the fitness function, and the weight factor is the characteristic weight determined according to target. At this time, the combination of optimization process parameters is the particle position (four-dimensional coordinates), and the output value of the weighted multi-objective fitness function is the particle fitness. To eliminate the dimension influence of different process parameters, performance indicators, process parameters, the performance index of the model involving all normalized processing, and the fitness function *F*(*x*) expression are as shown in Equation (6).
(6)$$F\left(x\right)=k_{1}D\left(x\right)+k_{2}\sigma_{b}\left(x\right)+k_{3}\sigma_{s}\left(x\right)+k_{4}A\left(x\right),\;x\in X$$

*x* is the particle coordinates’ vector, containing (*x*_1_, *x*_2_, *x*_3_, *x*_4_), indicating the laser power, scanning speed, scanning space, and spread powder process parameters with a thick layer of normalized values; *D*(*x*), *σb*(*x*), *σs*(*x*), *A*(*x*) are the density, tensile strength, yield strength, and break elongation of the normalized prediction model; *k*_1_, *k*_2_, *k*_3_, *k*_4_ are the four performance indicators according to the target to determine the weight initialization (when using the analytic hierarchy process (AHP) to determine the weight); and *X* is the particles in the search space constraints, namely the particle four-coordinate normalization process parameter value’s scope, which here are all [−1, 1].

(3) Determine the processing strategy of the particle crossing the boundary. The particle speed (*v*)’s scope is [−0.1, 0.1], when the speed cross-border *v* is the closest to the boundary value of the particle flying coordinate (*xi*)’s value in the range of [−1, 1] at the position of the cross-border to prevent particles trapped in the local optimum, with the coordinates *x_i_* calling back to 0.

(4) Determine the hyperparameters. The value range of learning factors $c_{1}$ and $c_{2}$ is set to 2. The number of population N is set to 1000, and the diversity of the population is increased to increase the global search ability. Because of the rapid convergence of the PSO, the maximum number of updates T only needs to be 100. The inertia factor *w*(*t*) is updated using linearly decreasing weights, making the global optimization strong at the beginning of the search and with strong local optimization when approaching global optimum, as shown in Equation (7) [25].
(7)$$w\left(t\right)=\left(w_{ini}-w_{end}\right)\left(T-t\right)/T+w_{end}$$

Among them, when the *w_ini_* for the initial inertia factor is 0.9, the *w_end_* inertial factor for the maximum update frequency is 0.4.

(5) End condition optimization. Usually the particle swarm optimization algorithm takes the maximum number of updates *T* as the end condition, but it usually converges before reaching the maximum number *T*. In order to accelerate the model optimization process, this paper proposes a heuristic end strategy combining targets: when the global optimal position of the particle swarm meets the target, the update iteration is considered as having found the optimal position, and the update iteration is ended in advance. The end condition of optimization can be expressed as follows:(8)$$\left\{\begin{array}{c} t\geq T\\ or\\ \;D\left(x\right)\geq z\;\;and\;\;\sigma_{b}\left(x\right)\geq l\;\;and\;\;\sigma_{s}\left(x\right)\geq q\;\;and\;\;A\left(x\right)\geq y \end{array}\right.$$
where *t*, *z*, *l*, *q*, and *y* are the density, tensile strength, yield strength, and elongation after break required.

The optimization objectives are densities of 100%, a tensile strength of 1100 MPa, a yield strength of 800 MPa, and a 30% elongation after break, and the optimization of the process parameters is carried out simultaneously with the standard particle swarm model using the weighted particle swarm model. Assuming a larger proportion of elongation after fracture, the ratio of each performance feature in the weighted PSO model is 0.168:0.168:0.168:0.496 calculated by the hierarchical analysis method, while each performance feature in the standard PSO model has an equal weight and is 0.25. As shown in Figure 5, compared to the standard PSO, the WPSO model is able to perform a faster and more accurate recommendation for SLM process parameters.

## 3. Experimental Section

BLT-S210 was used as SLM experimental equipment, the laser power of the equipment could reach 500 W, the scanning speed was 0~7 m/s, the lowest-forming layer thickness could reach 15 μm, and the process parameters could be adjusted in a wide range. The forming size of the device was 105 mm × 105 mm × 200 mm (excluding the substrate thickness). The equipment is shown in Figure 6. 

High-temperature alloy GH4169 was used as experimental material. GH4169 is a precipitation-strengthened nickel-based high-temperature alloy with good comprehensive properties in the temperature range of −253~650 °C. The main chemical compositions are shown in Table 7. The particle size of the high-temperature alloy GH4169 powder used in this experiment is normally distributed between 15 and 45 μm, and more than 90% of the powder particle sizes are about 44 μm, as shown in Figure 7.

The density was measured by drainage method. The measuring equipment was ATY124 electronic balance produced by Shimadzu in Japan, and the minimum dividing value was 0.1 mg. The measurement experiment diagram is shown in Figure 8.

The measuring instrument for tensile mechanical properties is the AG-IC 100 kN electronic universal testing machine produced by Shimadzu in Japan, as shown in Figure 9.

After the performance measurement was completed, combined with the results of previous exploration experiments, a total of 58 data records were collected from the current SLM process database for subsequent process knowledge base and process recommendation research.

## 4. Discussion

Upon inputting the target parameters, the weighted case-based reasoning model is employed to retrieve historical process cases. If similar cases meeting the similarity threshold and target criteria are identified, the corresponding case solution is output as the recommended result of the model. In cases in which no such matches are found, the weighted particle swarm optimization (WPSO) model engages in parameter optimization. The output process plan generated after the optimization is then suggested by the model. The accuracy of the process recommendation is defined as the level of correspondence between the features of the target and those of the formed parts following experimental machining based on the recommended process scheme. Given the multitude of features and their diverse dimensions, a unified index is essential for holistic measurements. Therefore, this paper proposes a method to calculate the process recommendation accuracy, as indicated by Equation (9).
(9)$$Ac=\overset{n}{\underset{i=1}{\sum }}\omega_{i}Ac_{i}$$

In this context, Ac denotes the accuracy of the process recommendation model, $\omega_{i}$ represents the normalized weight of the ith local feature, and $Ac_{i}$ signifies the recommended accuracy of the ith local feature.

As practical processing places greater emphasis on performance requirements compared to other demands, this study confines the calculation of the model’s process recommendation accuracy to the performance requirement’s intrinsic characteristics. This includes the density, tensile strength, yield strength, and elongation at break. The formula for calculating the recommended accuracy of local features is presented in Equation (10).
(10)$$Ac_{i}=\left\{\;\;\begin{array}{c} \frac{n_{i}}{N_{i}}\times 100\%,\;\;\;\;\;\;\;n_{i}<N_{i}\\ 100\%,\;\;\;\;\;\;\;\;\;\;n_{i}\geq N_{i} \end{array}\right.$$

Here, $Ac_{i}$ symbolizes the accuracy of the ith local feature, with *n_i_* indicating the measured value of the ith local feature after processing using the recommended technological scheme and *N_i_* representing the target parameter of the *i*th local feature. As all the local features involved in the calculation are performance parameters, a measurement value exceeding the target is considered a complete match, yielding a local feature recommendation accuracy of 100%. The experimental results are presented below.

Experiment 1: When the target is basically similar to cases in the case base.

We use a simulation to generate the target as the experimental scheme for the “basically similar” case, and the target for Experiment 1 is shown in Table 8.

When the target is similar to the cases in the case library, the AHP–WPSO model searches for cases that meet the similarity threshold and match the target for the process recommendation. The process solutions are shown in Table 9. Under the condition that the similarity of all the recommended process solutions meets the threshold, the solution of the case from the corresponding similar cases is extracted as the process solution for the processing test, and the measured performance results after processing are shown in Table 10.

As shown in Figure 10, except for the recommended accuracy of test case No.3, which is 100%, the recommended process accuracy of the remaining test cases is slightly lower than 100%, but at a high level. By comparing this with the target in Table 8, it can be seen that the tensile strength after the processing of the recommended process solution in case No.1 is about 15 MPa lower than the target, the elongation after break after the processing of the recommended process solution in No.2 is 0.4% lower than the target, the denseness after the processing of the recommended process solution in No.4 is 0.1% lower than the target, the denseness after the processing of the recommended process solution in No.5 is 0.4% lower than the target, and the elongation after break is 0.4% lower than the target. The elongation is 0.1% lower than the target. The comparison results show that there is an error in the recommendation results of the hybrid case reasoning/particle swarm model for “basically similar” cases, but the error is controlled at a low level. Overall, the average process recommendation accuracy of the hybrid case inference/particle swarm model reaches 99.81% when there are cases in the case library that are similar to the target.

Experiment 2: In order to ensure the authenticity and effectiveness of the experiment.

To ensure the real validity of the experiment, the targets for Experiment 2 were formed by simulating the mechanical properties after extracting them from other papers [21,26,27], for a total of five test cases, as shown in Table 11.

If there are no similar cases in the case library, the hybrid case reasoning/particle swarm model generates process solutions through weighted particle swarm optimization and then recommends them. The process solutions recommended by the model are shown in Table 12.

As shown in Table 12 and Figure 11, it can be seen that the recommended accuracy of the process of test cases No.7, No.8, No.9, and No.10 is at a high level, and only the recommended accuracy of test case No. 6 is low compared to the target in Table 13. There are differences from the targets in terms of densities, tensile strengths, yield strengths, and post-break elongations in test case No.6. The tensile strengths and yield strengths of the target are too high, which exceed the range of the weighted particle swarm model. The post-break elongation of test case No.7 is 2.97% lower than the target, which may be affected by the accuracy of the prediction model in the weighted particle swarm model; test cases No.8 and No.9 have only a small difference between the densities and the target after processing, and the recommended process solutions of test cases No.8 and No.9 can meet the target under the consideration of test errors. There is no case similar to the target in the case library, and the average process recommendation accuracy of the case inference/particle swarm hybrid model reaches 96.32%.

When applying the hybrid case reasoning/particle swarm model for selective laser melting (SLM) process recommendations, our experimental tests and comparisons have revealed the presence of certain levels of errors. 

More specifically, by comparing the results, it is clear that the mixed case reasoning/particle swarm model exhibits a lower level of error in recommending solutions for “basically similar” scenarios. Its error may appear in the testing process. However, these errors are contained within a lower range. For instance, the densities, tensile strengths, yield strengths, and elongation at break of the processed results for Test Case 6 show significant deviations from the user’s requirements. This discrepancy may be attributed to excessively high user demands for tensile and yield strengths, surpassing the optimization scope of the weighted particle swarm model and leading to suboptimal outcomes. Test Case 7 shows an elongation at break that is 2.97% lower than the user’s requirement, possibly influenced by the precision of the predictive model within the weighted particle swarm model. In contrast, Test Cases 8 and 9 exhibit minor discrepancies only in density compared to the user requirements. Considering experimental errors, the recommended process schemes for Test Cases 8 and 9 meet the user demands.

## 5. Conclusions

This study introduces a novel SLM (selective laser melting) process recommendation approach employing the AHP–WPSO (analytic hierarchy process–weighted particle swarm optimization) model. The research aims to optimize four key process parameters in SLM fabrication, including laser power, scanning speed, scanning spacing, and powder layer thickness. The performance assessment criteria encompass compactness, tensile strength, yield strength, and the post-fracture elongation of the fabricated components. The validation of the proposed approach is conducted through experiments, leading to the following significant conclusions:The integration of the AHP–WPSO model effectively facilitates precise SLM process solution recommendations, incorporating essential parameters such as tensile strength, post-fracture elongation, and density. Despite minor variations observed in specific scenarios, the model consistently demonstrates a high level of accuracy in its recommendations, contributing to advancing process design methodologies.Our experimental results validate the robustness of the AHP–WPSO model, exhibiting an impressive accuracy rate of 99.81% in scenarios with analogous process instances. Moreover, the model achieves a commendable accuracy rate of 96.32% in cases in which similar instances are lacking, highlighting its adaptability and reliability.

It is worth noting that while the AHP–WPSO methodology minimizes reliance on existing process knowledge, it possesses limitations in interpreting qualitative knowledge and handling qualitative data. Furthermore, the model has yet to address the challenges of complex parameter optimization, such as achieving a uniform residual stress distribution and an even distribution of alloy elements. Additionally, the metallurgical behavior of metals is not taken into consideration during the forming process. Future research efforts could enhance the model’s capabilities by integrating rule-based reasoning and expert inference methods, while also incorporating thermodynamic principles to establish metallurgical process models [28]. This approach can effectively tackle these challenges and further enhance the efficacy of the process recommendation framework.

The proposed AHP–WPSO model offers a robust and effective solution for SLM process recommendations, contributing significantly to the precision of process optimization and the overall enhancement of process design outcomes.

## Figures and Tables

**Figure 1 materials-16-05656-f001:**
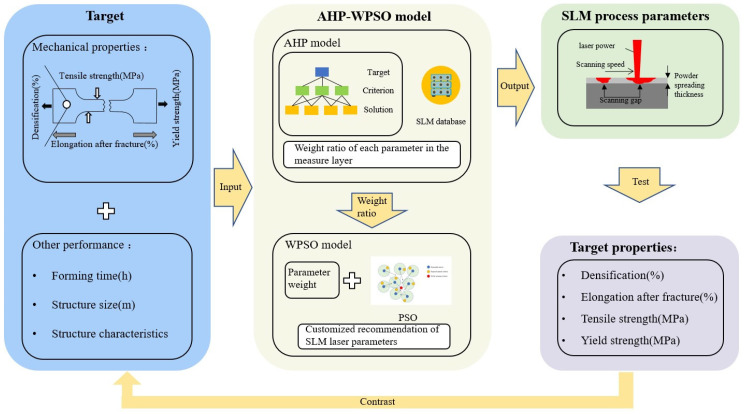
SLM process recommendation AHP–WPSO mixture model.

**Figure 2 materials-16-05656-f002:**
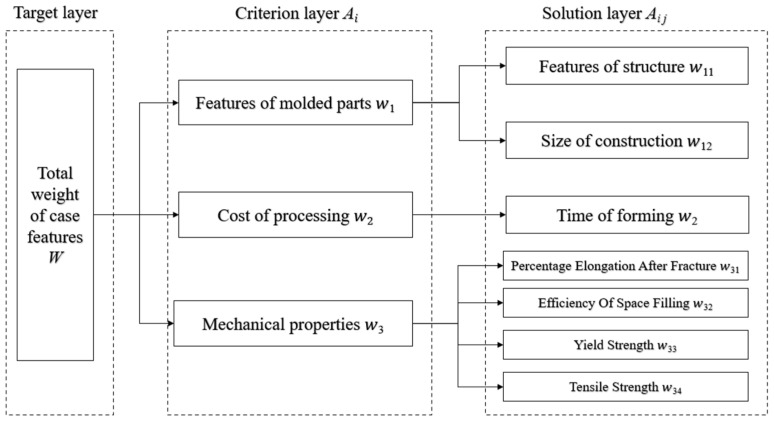
Schematic diagram of the case feature hierarchy model.

**Figure 3 materials-16-05656-f003:**
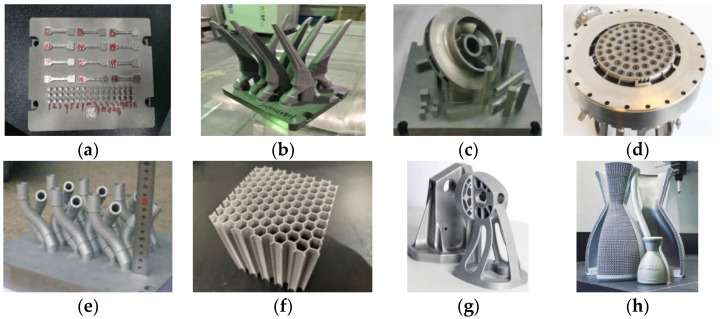
Physical diagram of typical structural features [23,24]: (**a**) Ordinary bulk block; (**b**) Complex sharps; (**c**) Impeller Design; (**d**) Interlayer runner; (**e**) Irregular shaped pipe; (**f**) Honeycomb structure; (**g**) Complex shell; (**h**) Lattice structure.

**Figure 4 materials-16-05656-f004:**
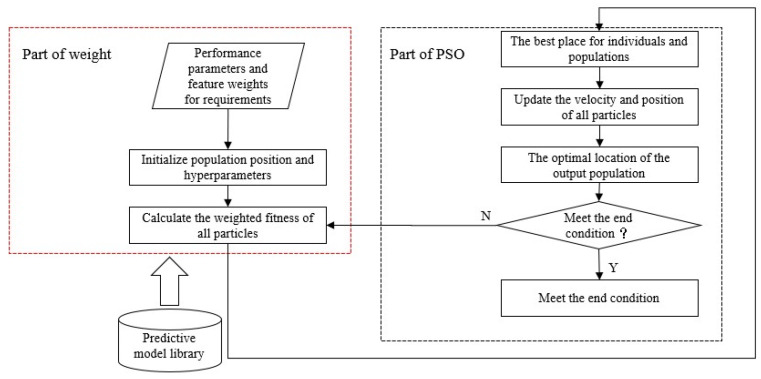
Weighted particle swarm optimization process.

**Figure 5 materials-16-05656-f005:**
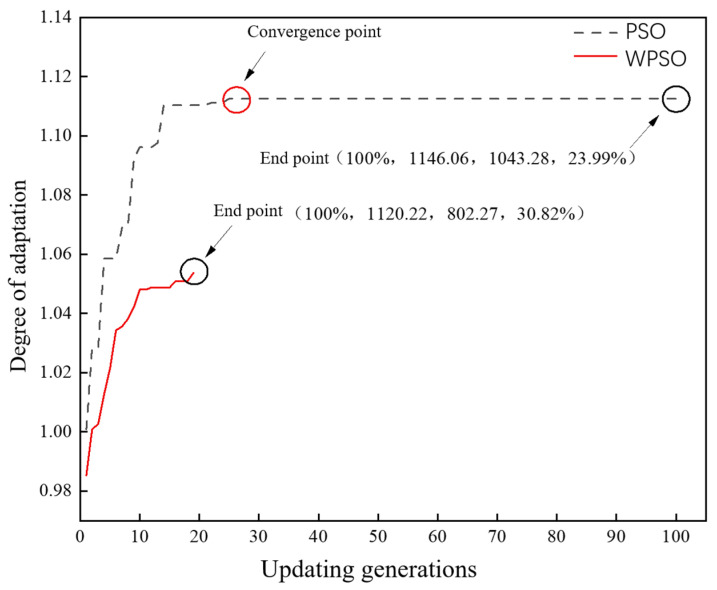
Iterative update of standard PSO and WPSO models.

**Figure 6 materials-16-05656-f006:**
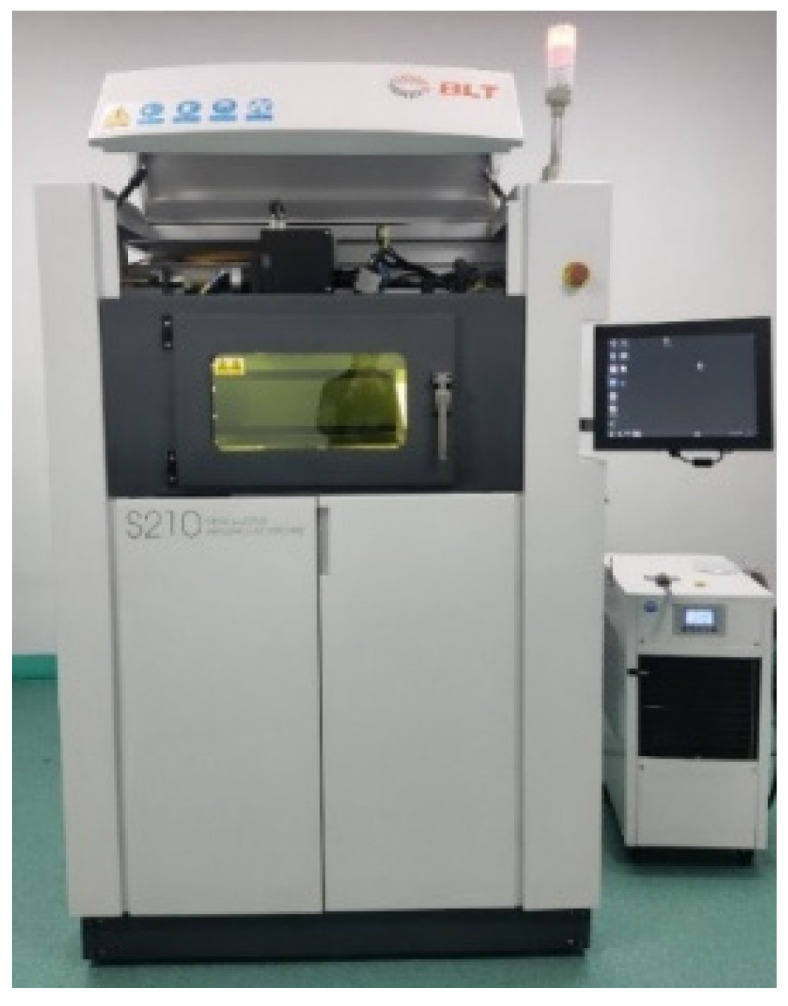
BLT-S210 SLM forming equipment.

**Figure 7 materials-16-05656-f007:**
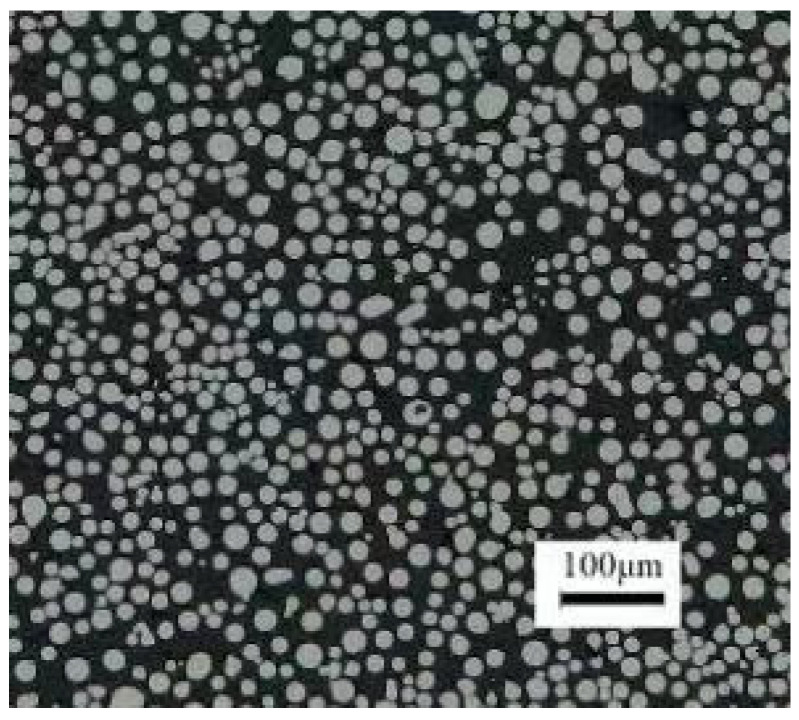
SEM image of GH6147 metal powder.

**Figure 8 materials-16-05656-f008:**
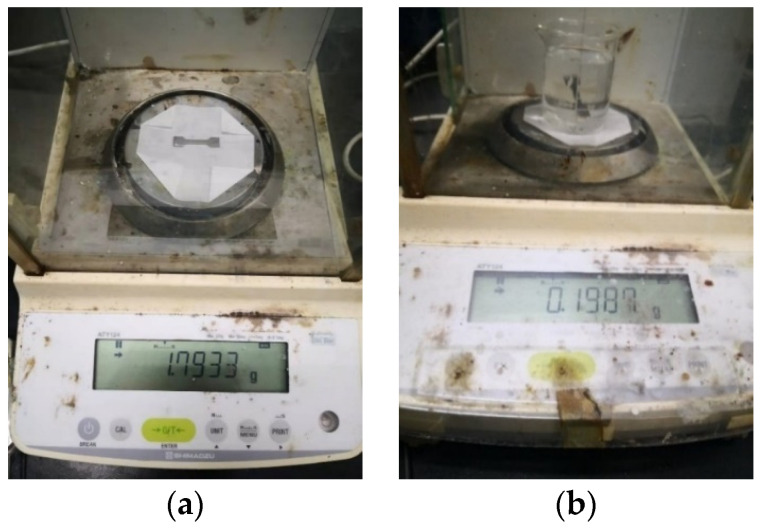
Schematic diagram of density measurement experiment: (**a**) Electronic balance to measure dry weight of formed parts; (**b**) Electronic balance to measure drainage weight.

**Figure 9 materials-16-05656-f009:**
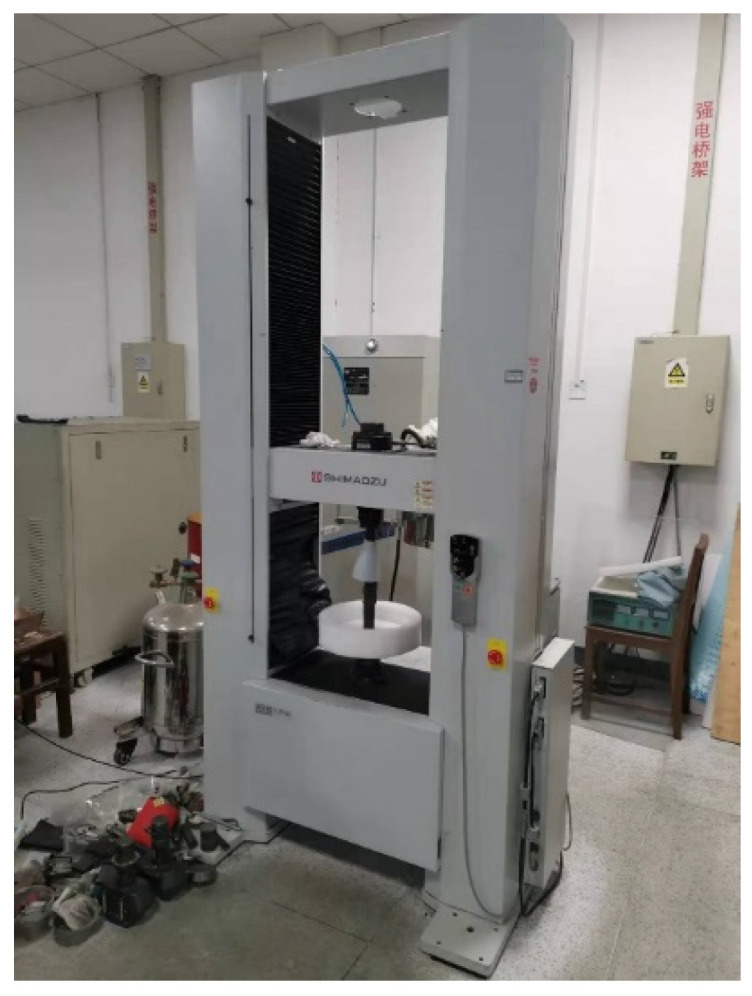
AG-IC 100 kN electronic universal testing machine.

**Figure 10 materials-16-05656-f010:**
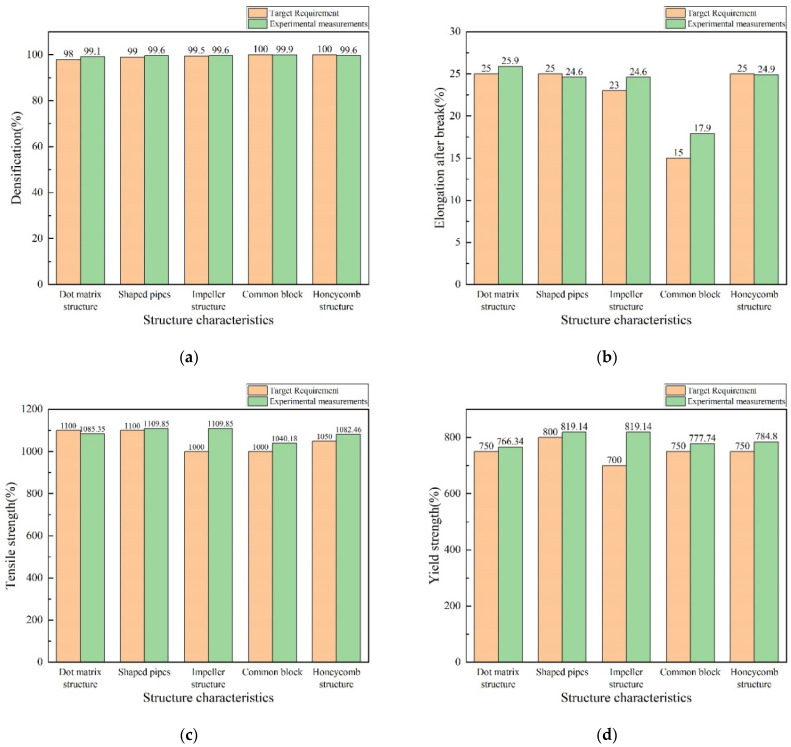
Performance comparison of target and experimental tests in Experiment 1: (**a**) dense density; (**b**) elongation at break; (**c**) yield strength; and (**d**) tensile strength.

**Figure 11 materials-16-05656-f011:**
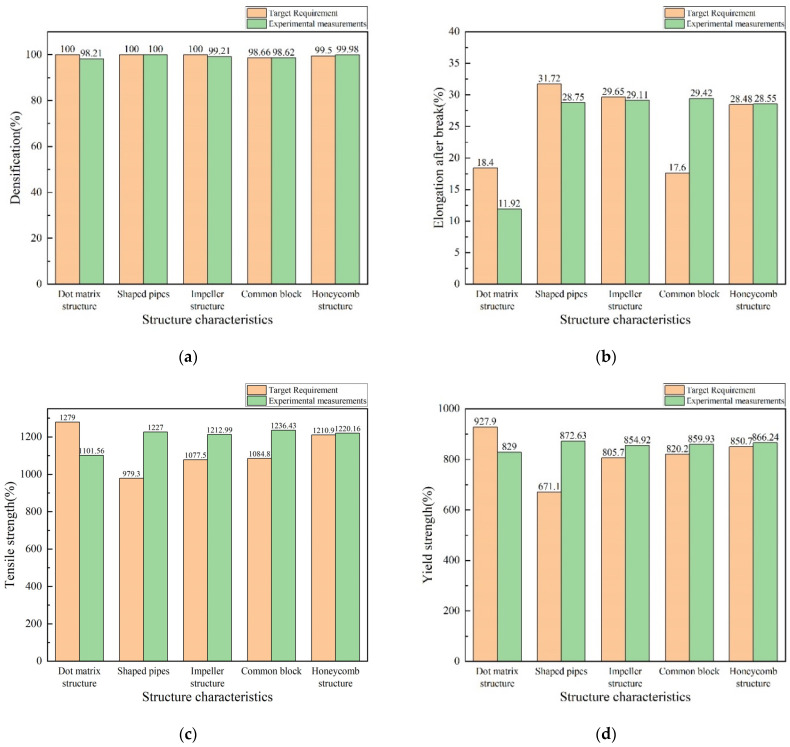
Performance comparison of target and experimental tests in Experiment 2: (**a**) dense density; (**b**) elongation at break; (**c**) yield strength; and (**d**) tensile strength.

**Table 1 materials-16-05656-t001:** Significance of the method’s scale from 1 to 9.

Scaling	The Meaning of the Scale
1	The features on the horizontal axis of the judgment matrix are equally important as the features on the vertical axis.
3	The features on the horizontal axis of the judgment matrix are slightly more important than the features on the vertical axis.
5	The features on the horizontal axis of the judgment matrix are more important than the features on the vertical axis.
7	The features on the horizontal axis of the judgment matrix are noticeably more important than the features on the vertical axis.
9	The features on the horizontal axis of the judgment matrix are significantly more important than the features on the vertical axis.
2, 4, 6, 8	Midpoint on the scale.

**Table 5 materials-16-05656-t005:** Average stochastic consistency index of judgment matrix of orders 1~10 [20].

Matrix Order	1	2	3	4	5	6	7	8	9	10
RI	0	0	0.58	0.90	1.12	1.24	1.32	1.41	1.45	1.49

**Table 6 materials-16-05656-t006:** Local similarity matrix of structural features.

Structure	Block	Angle	Impeller	Channel	Pipe	Honeycomb	Shell	Lattice
Block	1	0.2	0	0	0	0.4	0	0.1
Angle	0.2	1	0	0.1	0	0.6	0.1	0.2
Impeller	0	0	1	0.6	0.6	0.2	0	0.1
Channel	0	0.1	0.6	1	0.8	0	0.4	0.1
Pipe	0	0	0.6	0.8	1	0.2	0.5	0
Honeycomb	0.4	0.6	0.2	0	0.2	1	0.4	0.6
Shell	0	0.1	0	0.4	0.5	0.4	1	0.2
Lattice	0.1	0.2	0.1	0.1	0	0.6	0.2	1

**Table 7 materials-16-05656-t007:** The main chemical composition of high-temperature alloy GH4169.

Composition	Cr	Ni	Co	Mo	Nb	C	Fe
Request/%	17.0~21.0	50~55.0	≤1.0	2.80~3.30	4.75~5.50	≤0.08	Bal
Measurement/%	18.98	54.66	0.12	3.15	5.00	0.024	Bal

**Table 8 materials-16-05656-t008:** Target for Experiment 1.

Test Number	Material Grades	Structural Features	Structure Size/mm^3^	Forming Time/min	Densification/%	Tensile Strength/MPa	Yield Strength/MPa	Elongation after Break/%
1	GH4169	Dot matrix structure	330	60	98.00	1100.00	750.00	25.00
2	GH4169	Shaped pipes	315	75	99.00	1100.00	800.00	25.00
3	GH4169	Impeller structure	325	75	99.50	1000.00	700.00	23.00
4	GH4169	Common block	315	50	100.00	1000.00	750.00	15.00
5	GH4169	Honeycomb structure	305	50	100.00	1050.00	750.00	25.00

**Table 9 materials-16-05656-t009:** The process scheme recommended by the model in experiment 1.

Test Number	Case Number	Equipment Name	Laser Power/W	Scanning Speed/mm/s	Scan Spacing/μm	Powder Layer Thickness/μm	Scanning Method	Similarity
1	Particle	BLT-S210	150	900	60	40	Checkerboard shape	0.908
2	Particle	BLT-S210	260	1300	110	20	Checkerboard shape	0.902
3	Particle	BLT-S210	260	1300	110	20	Checkerboard shape	0.860
4	Particle	BLT-S210	350	600	150	20	Checkerboard shape	0.861
5	Particle	BLT-S210	290	1100	90	20	Checkerboard shape	0.908

**Table 10 materials-16-05656-t010:** Performance results measured after processing tests using the recommended process scheme in Experiment 1.

Test Number	Densification/%	Tensile Strength/MPa	Yield Strength/MPa	Elongation after Break/%	Accuracy Ac
#1	99.10	1085.35	766.34	25.90	99.67%
2	99.60	1109.85	819.14	24.60	99.60%
3	99.60	1109.85	819.14	24.60	100.0%
4	99.90	1040.18	777.74	17.90	99.98%
5	99.60	1082.46	784.80	24.90	99.80%

**Table 11 materials-16-05656-t011:** Target for Experiment 2.

Test Number	Material Grades	Structural Features	Structure Size/mm^3^	Forming Time/min	Densification/%	Tensile Strength/MPa	Yield Strength/MPa	Elongation after Break/%
6	GH4169	Common block	1373.75	90	100.00	1279.00	927.90	18.40
7	GH4169	Common block	1373.75	90	100.00	979.30	671.10	31.72
8	GH4169	Common block	1373.75	90	100.00	1077.50	805.70	29.65
9	GH4169	Common block	1373.75	100	98.66	1084.80	820.20	17.60
10	GH4169	Common block	1373.75	150	99.50	1210.90	850.70	28.48

**Table 12 materials-16-05656-t012:** The process scheme recommended by the model in experiment 2.

Test Number	Case Number	Equipment Name	Laser Power/W	Scanning Speed/mm/s	Scan Spacing/μm	Powder Layer Thickness/μm	Scanning Method	Similarity
6	Particle	BLT-S210	250	300	150	30	Checkerboard shape	——
7	Particle	BLT-S210	150	420	60	30	Checkerboard shape	——
8	Particle	BLT-S210	120	480	60	30	Checkerboard shape	——
9	Particle	BLT-S210	120	370	60	30	Checkerboard shape	——
10	Particle	BLT-S210	130	520	65	30	Checkerboard shape	——

Note: The weighted particle swarm model generates the process scheme, so there is no corresponding case number and the similarity cannot be calculated.

**Table 13 materials-16-05656-t013:** Performance results measured after processing tests using the recommended process scheme in Experiment 2.

Test Number	Densification/%	Tensile Strength/MPa	Yield Strength/MPa	Elongation after Break/%	Accuracy Ac
6	98.21	1101.56	829.00	11.92	84.62%
7	100.00	1227.00	872.63	28.75	97.66%
8	99.21	1212.99	854.92	29.11	99.35%
9	98.62	1236.43	859.93	29.42	99.99%
10	99.98	1220.16	866.24	28.55	100.00%

## Data Availability

Not applicable.

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
