# Peer review of "Recommendation of SLM Process Parameters Based on Analytic Hierarchy Process and Weighted Particle Swarm Optimization for High-Temperature Alloys"

_materials, 2023, doi:10.3390/ma16165656_

Round 1

Reviewer 1 Report

I have reviewed the manuscript titled "Recommendation of SLM process parameters based on analytic hierarchy process and weighted particle swarm optimization for high-temperature alloy" and I would like to recommend its publication in the journal. I enjoyed reading this article and I believe the idea proposed by authors is very interesting. As rightfully mentioned by authors, SLM is a very complicated process and setting an optimum working condition for such a complicated process is extremely difficult. The paper is well-organized and most details are well-interpreted. I suggest publication of this paper after following mandatory revisions:

1.    There are two extremely important issues when it comes to the optimization of SLM, and none are even mentioned in the manuscript. These issues include residual stress and homogeneity of alloying elements distribution. I am not saying that these two “complicated-to-measure” parameters should be part of your model. But, I believe they should be mentioned somewhere in your manuscript, perhaps as issues which need to be taken into considerations in future modelling approaches. I highly recommend these two manuscripts, as they address mentioned parameters.

-        Zhang, S., Han, B., Zhang, T., Chen, Y., Xie, J., Shen, Y., Huang, L., Qin, X., Wu, Y., Pu, K. (2023). High-temperature solid particle erosion characteristics and damage mechanism of AlxCoCrFeNiSi high-entropy alloy coatings prepared by laser cladding. Intermetallics, 159, 107939. doi: https://doi.org/10.1016/j.intermet.2023.107939

-        Zhu, Q., Chen, J., Gou, G., Chen, H., & Li, P. (2017). Ameliorated longitudinal critically refractedAttenuation velocity method for welding residual stress measurement. Journal of Materials Processing Technology, 246, 267-275. doi: https://doi.org/10.1016/j.jmatprotec.2017.03.022

2.    Please comment on the uncertainty and errors of predictions in your model.

3.    Please provide more information on the alloy used in your investigation.

4.    No need to add the picture of tensile machine.

5.    Images and charts are mostly very small with low quality.

6.    Conclusion should be written in a way that it gives an independent and understandable message. When you say “Sample No. 1” in the conclusion, it does not make any sense.  Please re-write the conclusion, having this point in mind.

7.    Also, it is good to mention somewhere the necessity of bringing metallurgical models into the future versions of these optimization algorithms. Please refer to the following article:

-        Kuang, W., Wang, H., Li, X., Zhang, J., Zhou, Q., Zhao, Y. (2018). Application of the thermodynamic extremal principle to diffusion-controlled phase transformations in Fe-C-X alloys: Modeling and applications. Acta Materialia, 159, 16-30. doi: https://doi.org/10.1016/j.actamat.2018.08.008

8.    The theoretical foundation of the optimization model should be explained in more details.

The English is recommended to be re-polished, though overall the paper is understandable and easy to follow. 

Reviewer 2 Report

A reference was given for the introduction's first sentence, but more references should be given in the following information.

There are significant grammatical errors in the study. A native English speaker should check it.

The quality of Figure 3 is poor.

More information should be given about the high-temperature material used in the study.

The last sentence of the first paragraph of the experimental work section is missing.

It will be valuable for the reader to provide SEM images of the powders used for SLM in the study.

More details should be given about the measurement of densities of materials. What is the theoretical density of the raw material, and how was it measured? A post-production microstructure image is not presented. Therefore, the density results are difficult to evaluate.

There are significant grammatical errors in the study. A native English speaker should check it.

Reviewer 3 Report

The authors wrote the recommendation of SLM process parameters based on analytic hierarchy process and weighted particle swarm optimization for high-temperature alloy. The manuscript had an interesting topic and was well-written, however, it could only be accepted with the following corrections:

1.     The abstract of the paper remains vague; a brief of summary was not highlighted clearly. What is SLM? At the beginning of the phrase, the full name should be stated.  

2.     SLM is one of the most widely used additive manufacturing (AM) processes for metallic materials. So, it is advised at the beginning of the Introduction Section the authors should define briefly about AM technology and its advantages. Therefore, it is recommended the authors can add the following papers as references:

·       Role of additive manufacturing and various reinforcements in MMCs related to biomedical applications. Advances in Materials and Processing Technologies (2022), 1-18.

·       Effect of HBN fillers on rheology property and surface microstructure of ABS extrudate. J. Teknol, 84, 175-182. https://doi.org/10.11113/jurnalteknologi.v84.16963

·       A review of natural fiber-based filaments for 3D printing: filament fabrication and characterization. Materials, 16(11), 4052. https://doi.org/10.3390/ma16114052

3.     Aim of the study or objective was not found at the end of the Introduction chapter.

4.     Refer to lines 95 and 103, please check the grammar (e.g. capital letter, incomplete sentence ‘Shown in Figure 2’, etc.)

5.     Y1, Y2, and Y3 ? It is better to define it properly.

6.     Line 113, “mechanical properties of formed parts>features of formed parts>processing cost”, What it is? ‘>’ greater than?

7.     Are the equations in lines 114, 116, 123, and 124? If yes, make them the proper equations.

8.     Refer to line 263, “as shown in……?” missing sentence.

9.     Figure 8(b), “Drawing of tensile specimen forming” 2D drawing or picture of specimens? Figure 8(c) should be a universal testing machine (UTM).

10.  Where is the Result and Discussion section?

11.  Paper lacks in writing the discussion of the result obtained and citing the previous studies.

12.  The conclusion could be improved by adding the future study, limitations, and implications for researchers.

some grammatical errors were found.

Round 2

Reviewer 1 Report

I would like thank authors for their efforts in revising the manuscript. The revised is acceptable. 

Reviewer 3 Report

All comments have been addressed by the authors.